# Comparison of Analytical Values D-Dimer, Glucose, Ferritin and C-Reactive Protein of Symptomatic and Asymptomatic COVID-19 Patients

**DOI:** 10.3390/ijerph19095354

**Published:** 2022-04-28

**Authors:** Nerea Pérez-García, Jessica García-González, Mar Requena-Mullor, Manuel Ángel Rodríguez-Maresca, Raquel Alarcón-Rodríguez

**Affiliations:** 1Torrecárdenas Hospital, 04009 Almería, Spain; perezgarcianerea11@gmail.com; 2Department of Nursing, Physiotherapy and Medicine, University of Almeria, 04120 Almería, Spain; mrm047@ual.es (M.R.-M.); ralarcon@ual.es (R.A.-R.); 3Clinical Laboratory Management Unit, Torrecárdenas Hospital, 04009 Almería, Spain; manuel.rodriguez.maresca.sspa@juntadeandalucia.es

**Keywords:** biomarkers, COVID-19, C-reactive protein, D-dimer, ferritins, glucose

## Abstract

Those infected by COVID-19 develop various kinds of complications with varying degrees of severity. For this reason, it is necessary to evaluate its analytical values to predict and reduce the risks and complications derived from this pathology. A cross-sectional study was carried out a population in Almeria (south-eastern Spain) who had a positive Polymerase Chain Reaction test result from 1 March 2020 to 30 November 2020. The study involved 4575 patients, with 1346 who were asymptomatic, 1653 mildly symptomatic (no hospitalisation needed) and 1576 severely symptomatic (symptomatic patients hospitalised). Laboratory values for D-dimer, glucose, serum ferritin, and C-reactive protein were analysed. The mean age of the participants in the study was 53.60 (16.89) years old. A total of 70.6% of the patients were symptomatic, of which 36.1% had mild symptoms. For all of the laboratory predictors analysed (D-dimer, glucose, serum ferritin, and C-reactive protein), it was found that severe alterations in the parameters were more frequent in severely symptomatic patients with statistically significant differences (*p* < 0.001), although these alterations also occurred in asymptomatic patients. Age correlated with analytical values (D-dimer, glucose, serum ferritin, and C-reactive protein) with statistically significant differences. Patients with severe symptoms presented alterations in the analytical values of D-dimer, glucose, serum ferritin, and C-reactive protein. Asymptomatic patients presented alterations in the analysed parameters, though with less frequency and severity than patients with severe symptoms.

## 1. Introduction

In December 2019, the SARS-CoV-2 virus appeared in Wuhan (China), causing the current pandemic known as Coronavirus-19 (COVID-19). SARS-CoV-2 mainly attacks the respiratory system, causing pneumonia with bilateral infiltrates that can progress to acute respiratory distress syndrome (ARDS), septic shock, multiple organ dysfunction syndrome, and can cause heart and kidney problems, coagulopathies, and even death [1]. Pathologies such as high blood pressure, cardiovascular diseases, coagulopathies, diabetes mellitus, and kidney failure are risk factors in terms of prognosis, evolution, and the appearance of complications [2,3,4,5]. For this reason, it is important to identify clinical and laboratory predictors that help to recognise severe forms of the disease [6]. Some of the most studied clinical markers are D-dimer, blood glucose, ferritin, C-reactive protein, platelet count, interleukin-6, creatinine, bilirubin, and fibrinogen [6].

Increased serum ferritin levels are associated with a worse prognosis, not only because of the relationship between elevated levels and the development of ARDS [1,6,7,8,9], but also the relationship of these levels with the most severe forms of the disease [10]. Elevated blood glucose values are also correlated with worse outcomes and higher mortality rates [3,5,11], regardless of whether the patient was diabetic before COVID-19 infection. In patients with elevated glucose levels, those with better control of glycaemic regulation showed a better prognosis than those with poor glycaemic control [12].

The appearance of ARDS, troponin-T elevation, and myocardial injury have been related to elevated levels of C-reactive protein [7]. Furthermore, Huang et al. [13] states that patients with high values of C-reactive protein have a higher risk of unfavourable evolution and increased mortality rates than those with standard values. D-dimer values (used to assess thrombotic events) have also been altered by COVID-19 infection [13]. Elevated D-dimer is a significant poor prognostic factor [14]. Several studies [4,6,15,16,17] suggest that in patients without D-dimer elevation, the incidence of venous thrombotic events is null, while in those who do present elevated values, the incidence is 50%. Given the high risk of developing thrombotic processes, it is vital to pay attention to asymptomatic patients as well [7]. 

Laboratory predictive values and baseline conditions that help predict the prognosis, development, and evolution of COVID-19 infection have been studied. However, it has been observed that the published literature focuses mainly on infected patients who develop symptoms, excluding those who are asymptomatic or develop mild symptoms that may be masked by other more common pathologies [7]. Therefore, the objective of this study was to analyse whether the analytical standards of serum ferritin, glucose, D-dimer, and C-reactive protein are altered in asymptomatic patients, as occurs in symptomatic patients infected with COVID-19.

## 2. Materials and Methods

### 2.1. Study Design

A cross-sectional study was carried out on a population in Almeria (in south-eastern Spain) who were affected by COVID-19 in order to analyse the analytical standards of serum ferritin, glucose, D-dimer, and C-reactive protein in asymptomatic and symptomatic patients.

### 2.2. Study Population and Data Collection

The study population consisted of 4575 patients from 18 to 80 years old with a positive COVID-19 result obtained through a Polymerase Chain Reaction (PCR) test, and had a blood test taken between the fifth and tenth day of infection. The day of infection was considered from the day of diagnosis of COVID-19 with positive PCR.

Participants were divided into two groups: asymptomatic and symptomatic. Within the symptomatic group, a subdivision was made between those who required hospital admission (considered to be severely symptomatic) and those who did not (considered to be mildly symptomatic). All patients were COVID-19 by confirmation with positive PCR. The patients were classified into three groups: (1) asymptomatic patients were those with positive PCR who had no symptoms but had been in close contact with other positive patients, (2) mildly symptomatic patients were patients with positive PCR who presented respiratory symptoms and who went to the emergency services as a result of these symptoms and received home follow-up, and (3) severely symptomatic patients, those who presented positive PCR and went to the emergency room due to respiratory distress and were admitted to the intensive care unit.

The patients in the study were recruited from the computer records in Diraya (a programme used by the Andalusian Health Service to maintain electronic medical records) of users affected by COVID-19 in the first five months of 2021. The data obtained were crossed with the records of blood test results analysed in the Torrecardenas Hospital Complex laboratory between 1 March and 30 November 2020. The inclusion criteria were patients older than 18 with a positive PCR test result who were willing to voluntarily participate in the study and patients not vaccinated against COVID-19. The exclusion criteria were patients who were minors and those who did not have a positive PCR test result and patients without a history of thrombotic disease.

None of the patients was immunized against COVID-19. The data was collected from 1 January to 31 May 2021, but the patients had been diagnosed with positive PCR from March 2020 to November 2020 before the start of the vaccination campaign in Almería (Spain).

The variables included in this study were sociodemographic variables (age and sex), comorbility (hypertension, type 2 diabetes mellitus, cardiovascular disease, chronic obstructive lung disease (COPD) and obesity), clinical variables (D-dimer, serum ferritin, glucose, and C-reactive protein) and patient symptoms (asymptomatic, mildly symptomatic, and severely symptomatic). In severely symptomatic patients, the mean number of days of hospitalization was also collected.

### 2.3. Data Analysis

To perform the statistical analysis, a database was created with the collected variables in the computer programme SPSS version 25.0 (Chicago, IL, USA). For the qualitative variables, frequencies were calculated with their corresponding percentages, and for quantitative variables, means, and standard deviations were calculated. For the comparison of qualitative variables, the Chi-square test (χ^2^) was used, considering a value of *p* < 0.05 as significant. For the comparison of means after a normality test (Kolmogorov-Smirnov test), the Kruskal-Wallis test was used. Quantitative variables were correlated through Spearman’s Rho coefficient. 

### 2.4. Ethical Considerations

Approval for this study was obtained from the Ethics and Research Commission of the Torrecardenas Hospital Complex of Almeria (COVID_21_URG_AP). All of the procedures were performed in accordance with the ethical standards of the Helsinki Declaration. The participants were informed about the study and their verbal and written consent was obtained.

## 3. Results

The mean age of the participants in the study was 53.60 (16.89), and 58.5% of them were female. Symptoms were developed in 70.6% of the patients, of which 36.1% presented mild symptoms. The patients with severe symptoms were the oldest participants (62.48 (15.21)). The patients with mild symptoms had a mean age of 51.70 (16.34). The asymptomatic patients were the youngest (47.81 (16.64)). A total of 62.2% of the patients were afflicted with at least one comorbid condition, including hypertension, type 2 diabetes mellitus, cardiac disease, COPD and obesity. The percentage of hypertension, type 2 diabetes mellitus, cardiovascular disease and obesity was similar in asymptomatic, mildly and severely symptomatic patients, with no statistically significant differences observed. The percentage of COPD was higher in severely symptomatic patients (3%) compared to mildly symptomatic and asymptomatic patients (1.3%, 1.1% respectively), these results being statistically significant (*p* = 0.04). (Table 1).

All patients with severe symptoms were hospitalized in the intensive care unit. The average length of hospital stay was 9.03 (5.32) days, with a range of 5–32 days. 

Table 2 shows the mean values of the analytical parameters that can be altered during the period of infection by COVID-19 in asymptomatic patients and in those with mild and severe symptoms.

Both males and females with severe symptoms presented the highest values of D-dimer, blood glucose levels, serum ferritin levels, and C-reactive protein, while the lowest values of these parameters were observed in asymptomatic males and females. The exception is for D-dimer values, where asymptomatic women have higher D-dimer values than mildly symptomatic women.

Statistically significant differences were observed when comparing the mean values of D-dimer, glucose, serum ferritin, and C-reactive protein between asymptomatic, mildly symptomatic, and severely symptomatic patients, both in males and females.

In asymptomatic females, mean D-dimer values were much higher than in asymptomatic males (females: 2139.12 (923.67) ng/mL; males: 410.80 (221.20) ng/mL). C-reactive protein levels were also slightly higher in asymptomatic females in relation to asymptomatic males (females: 0.62 (0.30) mg/dL, males: 0.53 (0.19) mg/dL). The rest of the analytical parameters, regardless of the symptoms presented, had higher values in males than in females. All of these results were statistically significant.

The comparison of the range of values between symptomatic and asymptomatic patients of the clinical parameters (D-dimer, glucose, serum ferritin and C-reactive protein) can be seen in Figure 1.

When analysing the alteration level of the analytical parameters in relation to the symptoms presented by the COVID-19 positive patients (Table 3), it was observed that the highest percentage of severe alterations of the values for D-dimer and C-reactive protein were seen in patients with severe symptoms (D-dimer: 44.7%; C-reactive protein: 46.5%). Asymptomatic patients and those with mild symptoms mostly presented normal values for both D-dimer and C-reactive protein. 

Hyperglycaemia was presented in 38.8% of the patients with severe symptoms, and 20.8% had mild alterations in blood glucose levels. Normal blood glucose levels were seen in 77.4% of the asymptomatic patients and 67.5% of those with mild symptoms.

Altered serum ferritin levels were presented in 78% of the males with severe symptoms and 69.4% of the females with severe symptoms. Serum ferritin values were mostly normal in both males and females with mild or asymptomatic symptoms. 

The correlation between age and analytical values (D-dimer, glucose, serum ferritin, and C-reactive protein) showed statistically significant results (Spearman’s Rho D-dimer: 0.52, *p* < 0.001; Spearman’s Rho glucose: 0.43, *p* < 0.001; Spearman’s Rho serum ferritin: 0.39, *p* < 0.001; Spearman’s Rho C-reactive protein: 0.37, *p* < 0.001). 

## 4. Discussion

The results obtained in this study demonstrate that patients with more severe symptoms presented alterations to a greater or lesser extent in the analytical values of D-dimer, glucose, serum ferritin, and C-reactive protein. There were also cases in which asymptomatic patients presented these laboratory abnormalities, although in smaller numbers. Biochemical follow-up of patients infected with SARS-CoV-2 is essential to assess the severity and progression of the disease. There are abundant studies that analyse the behaviour of laboratory markers in patients hospitalised for COVID-19 [6,16,18,19,20,21,22,23,24], some of them coinciding with the markers analysed in this study: D-dimer, glucose, serum ferritin, and C-reactive protein. A total of 67.5% of the patients were afflicted with at least one comorbid condition, including hypertension, type 2 diabetes mellitus, cardiac disease, COPD and obesity. These results are similar to those found in other studies [25,26].

### 4.1. D-Dimer Marker Analysis

The latest studies have considered D-dimer to be a good prognostic indicator of this disease, as it has been shown that patients infected with COVID-19 present higher numbers than healthy patients and that the higher the figure, the worse the prognosis and evolution [6,15,23,27]. Healthy participants were not included in this study; however, the same relationship was found in symptomatic and asymptomatic patients, with higher figures in severely symptomatic patients as in the aforementioned studies. Also coinciding with the results in this study is research that suggests that D-dimer levels greater than 1000 ng/mL represent an independent risk factor for poor prognosis [24]. Along these same lines, in the bibliographic review presented by Görlinger et al. [4], it was concluded that D-dimer differentiates severe patients from mild patients, and that a relationship exists between hospital death at an advanced age and values ≥ 1 µg/mL at admission. D-dimer values ≥ 2.0 µg/mL within the first 24 h of hospital admission are a good predictor of in-hospital mortality [4]. 

One of the complications of COVID-19 infection is the appearance of coagulation and thrombogenesis [23,28], as observed in a multicentre retrospective study that analysed the rate and severity of thrombotic complications in 400 hospitalised COVID-19 patients (144 in ICU). Elevated D-dimer levels at initial evaluation were found to be predictive of coagulation-associated complications during hospitalisation [27]. In this study, it was observed that patients with higher levels of D-dimer were patients with severe symptoms and required hospital admission. The risk of experiencing a thrombotic event was 6.79 [CI 95%, 2.39–19.30] times higher in patients with D-dimer levels greater than 2500 ng/mL. However, the risk of bleeding, although high, was lower (3.56 [CI 95%, 1.01–12.66]) [27]. Similar conclusions were drawn by another study in which D-dimer elevations above 243 ng/mL were detected in 63% of the patients. In this case, the mean D-dimer was calculated to be 3144.50 ± 1709.4 for hospitalised patients with severe pneumonia in the intensive care unit [29]. As in this study and studies previously mentioned, D-dimer values showed positive correlations with age, length of stay, and mortality [14,29].

In short, it can be assumed that the risk of developing severe complications, as well as the risk of mortality, is greater in those patients with D-dimer alterations than in those without [14,30,31]. Moreover, those infected with COVID-19, whether or not they required hospital admission, have a high risk of venous thromboembolism and therefore prophylactic treatment with low molecular weight heparins is recommended [7]. 

### 4.2. Glucose Marker Analysis

In this study, it was observed that the increase in glucose levels was positively related to age. Similarly, in the study by Want et al. [21], risk factors were identified for disease progression and mortality. The sample consisted of 2433 COVID-19 patients, of which 1758 were considered mild or moderate cases, and 675 severe or critical at hospital admission. Among those considered mild, 74 progressed to a severe or critical stage, and 41 of those labelled as severe or critical died [21]. Elevated blood glucose levels are negatively related to the severity of the disease [5], establishing relationships between elevated glucose levels with more medical interventions and increased risk of mortality in patients with COVID-19 and pre-existing, type-2 diabetes mellitus [32]. In this study, a relationship was observed between the increase in glucose and the development of more severe symptoms.

Various studies have confirmed that SARS-CoV-2 infection causes hyperglycaemia in people without pre-existing diabetes, exacerbating the condition in diabetic patients prior to infection [33]. Therefore, glycaemic control is important in any patient with COVID-19, whether or not there is a history of diabetes mellitus [3,24,32,34]. In addition, COVID-19 can increase the risk of hyperglycaemia and other complications in patients with or without a history of diabetes [35]. In this study, it was observed that some patients also suffered glucose alterations leading to hypoglycaemia. These results confirm the importance of monitoring blood glucose levels in COVID-19 patients with or without a previous diagnosis of diabetes mellitus.

### 4.3. Serum Ferritin Marker Analysis

Serum ferritin functions as a prognostic marker and forewarns of the progression to more severe forms of the disease [6,22]. In severe COVID-19, a hyperinflammatory state occurs with increased ferritin, which is associated with increased mortality rates, multiple organ dysfunction, and the need for admission into intensive care units [8,18,24,36,37]. Contrary to this idea, other authors suggest that ferritin is not a good predictor of mortality, stating that the value obtained when analysing the area below the ROC curve evidenced a low ability to distinguish predictions of mortality. The same occurred when the negative predictive value and positive predictive value were analysed [38]. Bolondi et al. [39] considers ferritin to be an early, nonspecific marker that rises in the first phase of the disease and takes about a month to fall. Therefore, it could be useful to distinguish patients who potentially require hospital admission, but it is not useful in patients who are already admitted into the ICU [39].

In the case-control study developed by Zhou et al. [40], 100 people participated who presented higher ferritin levels in the cases (COVID-19 patients) than in the controls (patients without COVID-19). In turn, these were higher in patients considered to be more severe, as were the results found in this study. However, no significant differences in age or sex were detected in the COVID-19 population that was divided into severe patients (7 males and 5 females) and mild patients (22 males and 16 females) [40]. On the contrary, in another study in which a sample of 56 patients who died from COVID and 245 hospitalised patients was analysed, significant differences were shown in age and length of stay between both groups [9]. In this study, significant differences were found in terms of sex, and a correlation was shown between age and changes in serum ferritin. It also showed a direct relationship between non-survivors and elevated serum ferritin levels [9]. This same relationship was found in another study, in which ferritin values were analysed in 191 hospitalised patients (with a mean age of 56 and predominantly males), of whom 54 did not survive [25].

### 4.4. C-Reactive Protein Marker Analysis

The hyperinflammatory state that occurs after COVID-19 infection produces an increase in C-reactive protein values [24]. One study suggests that the risk of developing serious events increase in by 5% for each unit that increases the concentration of C-reactive protein in COVID-19 patients [41]. As with the rest of the laboratory parameters mentioned above, C-reactive protein can also be a guide for the prognosis and evolution of COVID-19 patients [24]. In the results obtained by Vaquero-Roncero et al. [26], 150 patients under 75 years old were observed with a Charlson mortality rate of less than or equal to six (half of them admitted to the ICU). The results indicated a higher risk index for patients admitted to the ICU with a higher SOFA (Sequential Organ Failure Assessment) score and increased values of C-reactive protein [26]. Another study suggests that patients who died of COVID-19 had levels ten times higher than those who managed to recover [42]. In this study, values in deceased patients have not been analysed, but it has been demonstrated that the more severe the symptoms, the higher the levels of C-reactive protein.

Compared with mildly symptomatic patients, the severe patients had much higher levels of C-reactive protein (43.8 [12.3–101.9] mg/L versus 12.1 [0.1–91.4] mg/L; *p* = 0.000) [41]. These data coincide with those presented in this study. A regression analysis showed that C-reactive protein was significantly associated with worsening conditions in patients with non-severe COVID-19, with an area under the curve of 0.844 (confidence interval of 95%, 0.761–0.926) and an optimal threshold value of 26.9 mg/L. [41]. In this study, it was also shown that patients with severe alterations in C-reactive protein levels were more frequently associated with having more severe symptoms. Coinciding with this study, a meta-analysis was found in which the results of 25 studies were analysed with a total of 5320 patients, where similar conclusions to those previously mentioned were reached [13]. 

### 4.5. Strengths and Limitations

One of the main limitations of this study was the analysis of only four laboratory markers, leaving out many others that can provide precise information on the evolution of COVID-19 and the possible complications that may develop. Also, the risk of developing thrombotic events, hyperglycaemia, sepsis, anaemia, or any complication that may arise from the elevation of the laboratory markers analysed was not studied. Furthermore, by obtaining values from only the first analysis after infection, the analysis does not cover those patients whose symptoms later became more severe. Consequently, the risk of mortality could not be analysed when the analysed parameters were elevated.

The main strength of this study lies in the fact that a sample similar to the general population was analysed, including asymptomatic patients who presented symptoms at varying degrees of severity. The time frame in which the pandemic is developing (in most countries it has already been possible to control the exponential trend in the number of infections) forces us to move towards controlling the complications that the population affected by this may develop especially if any of these could compromise your health or quality of life. Until now, most of the studies work with a symptomatic population, excluding the asymptomatic one, and thus assuming that this population, by not developing symptoms, does not develop complications or sequelae either. However, in daily clinical practice we have been observing a significant increase in people who have presented for adverse events associated with complications derived from COVID-19 infection. What is noteworthy about this health demand is that a large number of these patients have been asymptomatic, with no underlying pathology that justifies the appearance of these complications and who, therefore, have not required hospital admission, follow-up, or any treatment. It is evident, therefore, that it is essential to investigate the mechanisms that help us to outline typologies of patients susceptible to developing serious sequelae, regardless of the existence or not of symptoms. Having the possibility of early identification of those potentially vulnerable individuals, whose clinical course may change precipitously, is vital to minimize damage to health. Extensively monitoring changes in analytical markers allows us to anticipate risk situations and avoid sequelae and complications, thus being able to prescribe prophylactic treatments and/or carry out actions aimed at recovering previous baseline functional capacity. Anticipating these situations will help us gain the human and material resources that we must make available to those affected when they need them.

## 5. Conclusions

The patients with severe symptoms presented alterations in the analytical values of D-dimer, glucose, serum ferritin, and C-reactive protein. Alterations in all of these parameters were severe in patients with severe symptoms. Asymptomatic patients presented alterations in the analysed parameters, though with less frequency and severity than patients with severe symptoms. Furthermore, age correlates with D-dimer, glucose, serum ferritin, and C-reactive protein.

Analysing the biochemical parameters of patients infected with COVID-19 is essential to assess the severity and progression of the disease, enabling the prediction and prevention of possible complications derived from COVID-19. Likewise, it can help to select the susceptible population to receive prophylactic treatment.

## Figures and Tables

**Figure 1 ijerph-19-05354-f001:**
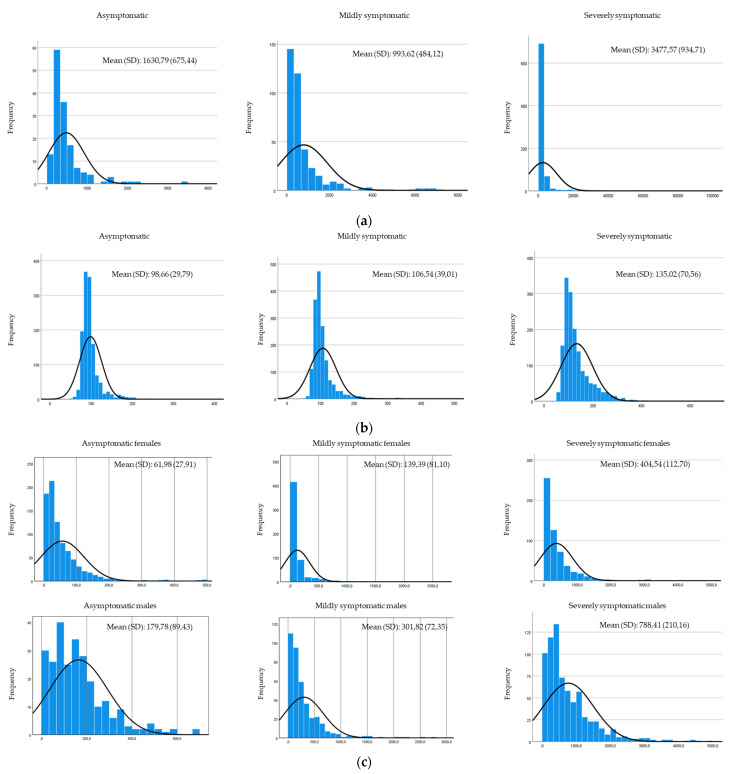
Comparison of the range clinical variables: (**a**) range of D-dimer values, (**b**) range of glucose values, (**c**) range of serum ferritin values, (**d**) range of c-reactive protein values.

**Table 1 ijerph-19-05354-t001:** Demographics and comorbidity characteristics of COVID-19 patients.

	Asymptomatic (*n* = 1346)	Mildly Symptomatic (*n* = 1653)	Severely Symptomatic (*n* = 1576)	*p-*Value ^a^
Age	47.81 (16.64)	51.70 (16.34)	62.48 (15.21)	0.001
Sex (female)	929 (59.2%)	924 (58.7%)	824 (57.5%)	0.59
Hypertension	201 (14.9%)	256 (15.4%)	233 (14.7%)	0.64
Type 2 Diabetes Mellitus	252 (18.7%)	324 (19.6%)	328 (20.8%)	0.22
Cardiovascular disease	168 (12.4%)	196 (11.8%)	187 (11.8%)	0.34
COPD	16 (1.1%)	23 (1.3%)	48 (3%)	0.04
Obesity	197 (14.6%)	217 (13.1%)	205 (13%)	0.19

(^a^
*p-*value obtained by Chi-square test).

**Table 2 ijerph-19-05354-t002:** Comparison of the analytical parameters’ mean values based on the patient’s symptoms and sex.

		D-Dimer ^b^	*p-*Value ^a^	Glucose ^c^	*p-*Value ^a^	Serum Ferritin ^b^	*p-*Value ^a^	C-Reactive Protein ^c^	*p-*Value ^a^
Males (*n* = 1898)	Asymptomatic	410.80 (221.20)	<0.001	104.97 (40.52)	<0.001	179.78 (89.43)	<0.001	0.53 (0.19)	<0.001
	Mildly symptomatic	1263.10 (621.04)		110.92 (41.98)		301.82 (72.35)		1.91 (0.76)	
	Severely symptomatic	4042.28 (1028.04)		136.48 (65.31)		788.41 (210.16)		6.75 (4.20)	
Females (*n* = 2677)	Asymptomatic	2139.12 (923.67)	<0.001	96.58 (24.95)	<0.001	61.98 (27.91)	<0.001	0.62 (0.30)	<0.001
	Mildly symptomatic	771.18 (331.34)		103.46 (36.49)		139.39 (81.10)		1.30 (0.98)	
	Severely symptomatic	2619.01 (813.44)		133.18 (76.69)		404.54 (112.70)		5.98 (3.54)	

(^a^
*p*-value obtained by Kruskal Wallis)/(^b^ Expressed in ng/mL)/(^c^ Expressed in mg/dL).

**Table 3 ijerph-19-05354-t003:** Distribution of the alteration level of the analytical values in relation to the symptoms presented by the patient.

			Asymptomatic (*n* = 1346)	Mildly Symptomatic (*n* = 1653)	Severely Symptomatic (*n* = 1576)	*p-*Value ^a^
**D-dimer ^b^**		**Normal** (≤500 ng/mL)	108 (70.6%)	226 (58.4%)	235 (29.2%)	<0.001
		**Mild** (501–1000 ng/mL)	29 (19.0%)	81 (20.9%)	210 (26.1%)	
		**Severe** (>1001 ng/mL)	16 (10.5%)	80 (2.7%)	359 (44.7%)	
**Glucose ^c^**		**Normal** (74–106 mg/dL)	1036 (77.4%)	1116 (67.5%)	564 (35.8%)	<0.001
		**Mild** (107–125 mg/dL)	136 (10.2%)	232 (14.0%)	328 (20.8%)	
		**Hyperglycaemia** (≥126 mg/dL)	120 (9.0%)	251 (15.2%)	612 (38.8%)	
		**Hypoglycaemia** (<74 mg/dL)	47 (3.5%)	54 (3.3%)	72 (4.6%)	
**Serum** **Ferritin ^b^**	**Male**	**Normal** (20–250 ng/mL)	188 (77.0%)	231 (61.3%)	160 (22.0%)	<0.001
		**Altered** (>250 ng/mL)	56 (23.0%)	146 (38.7%)	568 (78.0%)	
	**Female**	**Normal** (10–120 ng/mL)	658 (87.5%)	375 (70.4%)	166 (30.6%)	<0.001
		**Altered** (>120 ng/mL)	94 (12.5%)	158 (29.6%)	377 (69.4%)	
**C-reactive protein ^c^**		**Normal** (<0.5 mg/dL)	132 (82.5%)	360 (59.7%)	205 (23.4%)	<0.001
		**Mild** (0.6–1 mg/dL)	13 (8.1%)	74 (12.3%)	71(8.1%)	
		**Moderate** (1.1–4 mg/dL)	12 (7.5%)	92 (15.3%)	192 (21.9%)	
		**Severe** (>4 mg/dL)	3 (1.9%)	77 (12.8%)	407 (46.5%)	

(^a^
*p-*value obtained by Chi-square test)/(^b^ Expressed in ng/mL)/(^c^ Expressed in mg/dL).

## Data Availability

Data of this study are stored in an SPSS software Project.

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
