# Peer review of "Comparison of Analytical Values D-Dimer, Glucose, Ferritin and C-Reactive Protein of Symptomatic and Asymptomatic COVID-19 Patients"

_ijerph, 2022, doi:10.3390/ijerph19095354_

Round 1

Reviewer 1 Report

I have some conserns about the manuscript.

Major concern

1.Authors should note the novelty of the present study if they want to categorize the manuscript as an original article, but not review article. 

2.The scale of histogram was too large. The present form could not add the further infromation. The graph would help to make readers understood the difference of values among asymptomatic/mildly symptomatic/severely-.

Minor concern

1.Label of FIgure 1 of (a) and (b) was inversed. 

Author Response

REVIEWER 1

RESPONSE: We appreciate the suggestions of reviewer 1

Major concern

1.Authors should note the novelty of the present study if they want to categorize the manuscript as an original article, but not review article.

RESPONSE: The main strength and novelty of the study:

The time frame in which the pandemic is developing (in most countries it has already been possible to control the exponential trend in the number of infections) forces us to move towards controlling the complications that the population affected by this may develop. viruses, especially if any of these could compromise your health or quality of life.

Until now, most of the studies work with a symptomatic population, excluding the asymptomatic and, thus, assuming that this population, by not developing symptoms, does not develop complications or sequelae either.

However, in daily clinical practice we have been observing a significant increase in people who have consulted for adverse events associated with complications derived from covid-19 infection. What is noteworthy about this health demand is that a large number of these patients have been asymptomatic, with no underlying pathology that justifies the appearance of these complications and who, therefore, have not required hospital admission, follow-up, or any treatment.

It is evident, therefore, that it is essential to investigate the mechanisms that help us to outline typologies of patients susceptible to developing serious sequelae, regardless of the existence or not of symptoms.

Having the possibility of early identification of those potentially vulnerable individuals, whose clinical course may change precipitously, is vital to minimize damage to health. Extensively monitoring changes in analytical markers allows us to anticipate risk situations and avoid sequelae and complications, thus being able to prescribe prophylactic treatments and/or carry out actions aimed at recovering previous baseline functional capacity. Anticipating these situations will help us gain the human and material resources that we must make available to those affected when they need them.

2.The scale of histogram was too large. The present form could not add the further infromation. The graph would help to make readers understood the difference of values among asymptomatic/mildly symptomatic/severely-.

RESPONSE: Histograms in the manuscript are modified.

Minor concern

1.Label of Figure 1 of (a) and (b) was inversed

RESPONSE: It is modified in the manuscript.

Reviewer 2 Report

The authors have improved the previous version of the manuscript after reviewers’ comments and indications, mostly in the presentation of the results obtained.

However, the main strength of the study according to the authors - the inclusion of asymptomatic COVID-19 patients - is still not sufficiently emphasised in the discussion.

Furthermore, the reviewer is still not sure about the novelty of the results communicated in this work since several manuscripts communicating the predictive function of these biomarkers have been published, and the biomarkers analysed are used as predictors in clinical practice.

Author Response

REVIEWER 2

The authors have improved the previous version of the manuscript after reviewers’ comments and indications, mostly in the presentation of the results obtained.

RESPONSE: We appreciate the suggestions of reviewer 2

However, the main strength of the study according to the authors - the inclusion of asymptomatic COVID-19 patients - is still not sufficiently emphasised in the discussion. Furthermore, the reviewer is still not sure about the novelty of the results communicated in this work since several manuscripts communicating the predictive function of these biomarkers have been published, and the biomarkers analysed are used as predictors in clinical practice.

RESPONSE: The main strength and novelty of the study:

The time frame in which the pandemic is developing (in most countries it has already been possible to control the exponential trend in the number of infections) forces us to move towards controlling the complications that the population affected by this may develop. viruses, especially if any of these could compromise your health or quality of life.

Until now, most of the studies work with a symptomatic population, excluding the asymptomatic and, thus, assuming that this population, by not developing symptoms, does not develop complications or sequelae either.

However, in daily clinical practice we have been observing a significant increase in people who have consulted for adverse events associated with complications derived from covid-19 infection. What is noteworthy about this health demand is that a large number of these patients have been asymptomatic, with no underlying pathology that justifies the appearance of these complications and who, therefore, have not required hospital admission, follow-up, or any treatment.

It is evident, therefore, that it is essential to investigate the mechanisms that help us to outline typologies of patients susceptible to developing serious sequelae, regardless of the existence or not of symptoms.

Having the possibility of early identification of those potentially vulnerable individuals, whose clinical course may change precipitously, is vital to minimize damage to health. Extensively monitoring changes in analytical markers allows us to anticipate risk situations and avoid sequelae and complications, thus being able to prescribe prophylactic treatments and/or carry out actions aimed at recovering previous baseline functional capacity. Anticipating these situations will help us gain the human and material resources that we must make available to those affected when they need them.

Reviewer 3 Report

The authors responded to my concerns. This version is much improved and deserves to be published in IJERPH

Author Response

REVIEWER 3

RESPONSE: Thank you for accepting the new version of the manuscript.

Round 2

Reviewer 1 Report

I have to admit author's effort.

Reviewer 2 Report

I find the author's response correct.

This manuscript is a resubmission of an earlier submission. The following is a list of the peer review reports and author responses from that submission.

Round 1

Reviewer 1 Report

Majors:

-Unclear study design and insufficiently powered study. It is unclear how healthy controls were recruited. It is also unclear on what criteria the authors used to divide their cohort into symptomatic and asymptomatic patients.

-As patients were recruited in the vaccination campaign or more than 1 year after the pandemic outbreak, it should be determined whether these individuals were immunized (due to previous infections or vaccination for instance).

-Limited news value: biochemical parameters as biomarkers of the severity of COVID-19 have already been documented by several studies.

-A lack of clinical information prevents the reader to draw solid conclusions... maybe the observed results are influenced by ethnicity and/or medication/comorbidity highly prevalent in hospitalized COVID-19 patients. Please provide additional demographic information and clinico-biological data, including race/ethnicity, duration of hospitalization... 

Reviewer 2 Report

Pérez-García and colleagues present a manuscript communicating a study where they have compared values of four biomarkers in PCR-positive COVID-19 patients, classified in three groups: asymptomatic, mildly symptomatic and severely symptomatic. They concluded that severe symptomatic patients showed alterations in these biomarkers’ values whereas asymptomatic patients presented alterations with less frequency and severity. Furthermore, they consider the biomarkers analysed are useful to assess the severity and progression of COVID-19 and to select patients for prophylactic treatment.

The reviewer is not sure about the novelty of the results communicated in this work since several manuscripts communicating the predictive function of these biomarkers have been published, and the biomarkers analysed are used as predictors in clinical practice. The authors point out that the literature published lacks information about asymptomatic COVID-19 patients, therefore they included them in their work. However, despite the inclusion of asymptomatic patients is one of the main strengths of the work, the results of the asymptomatic group are sightly discussed.

The reviewer considers the following clarifications and changes necessary to improve the quality of the presented manuscript:

  • L78: How do the authors determine the infection day for sampling?
  • L110-114: These data should be presented in a table summarising all the characteristics of the study population.
  • L124-129: Are these differences statistically significant?  The reviewer recommends including the p-value for each comparison presented in the manuscript.
  • Table 1: It is not clear what has been compared using the Kruskal-Wallis test to obtain the p-value
  • L119-120: The authors indicate that asymptomatic women always present the lowest values, but Table 1 shows that asymptomatic women present higher D-dimer values than the mildly symptomatic woman.
  • L136: “Asymptomatic patients with mild symptoms mostly presented normal values”; it makes no sense, may “Asymptomatic patients and mild symptoms mostly presented normal values for”?
  • L194: The authors communicate that glucose levels are related to age but no data nor statistical result is shown.
  • L247 “…results obtained by Vaquero-Roncero and colleagues or et al.

Reviewer 3 Report

Authors studies the prevalence of biomarkers in patients with COVID-19. As authors mentioned, the normal range of biomarkers in asymptomatic patients remained unclear. Authors addressed the issue to evaluate a large number of patients. However, the manuscript suffers several problems.

Major concern

  1. The information of comorbidities which could affect the result such as diabetes, hematologic diseases, and heart diseases is lacking. Authors should add the general items regarding patient demographics.
  2. Hence the data was collected in the first half of 2021, the population would include the patients who had been vaccinated. The value of D-dimer could affect the vaccination. Authors should add the information of vaccination status (none/1 shoot/2 shoots).
  3. The novelty of the study is unclear. What distinct the manuscript from other studies? If authors want to stress the impact of the values in asymptomatic patients, authors should precisely compare the range of values between symptomatic patients and asymptomatic using histograms. Many papers have been reported patients with high-severity had higher value of biomarkers which was evaluated in the present study.